# Bacteria-mediated green synthesis of silver nanoparticles and their antifungal potentials against *Aspergillus flavus*

Achyut Ashokrao Bharose[1], Sunil Tulshiram Hajare[2]*, Gajera H. P.[3], Mukesh Soni[4], Krushna Kant Prajapati[5], Suresh Chandra Singh[6], Vijay Upadhye[7]

**1** Marathawada Agriculture University, Parbhani, M. S., India, **2** College of Natural and Computational Sciences, Dilla University, Dilla, Ethiopia, **3** Department of Biotechnology, College of Agriculture, Junagadh Agricultural University, Junagadh, Gujarat, India, **4** Dr D.Y Patil Vidyapeeth, Pune and Department of CSE, University Centre for Research & Development Chandigarh University, Mohali, Punjab, India, **5** Shri Krishna Pathology & ELISA Laboratory, Sumerpur, India, **6** Bilva Laboratories Pvt Ltd, BHU, Varanasi, India, **7** Institute of Applied Sciences, Research & Development Cell, Parul University, Vadodara, Gujarat, India

* sunilhajare@gmail.com

**Data Availability Statement:** All data discovered or designed in this study is available in this article and in Additional files. Also, the datasets generated during the current study are available in nucleotide

## Abstract

The best biocontroller *Bacillus subtilis* produced silver nanoparticles (AgNPs) with a spherical form and a 62 nm size through green synthesis. Using UV-vis spectroscopy, PSA, and zeta potential analysis, scanning electron microscopy, and Fourier transform infrared spectroscopy, the properties of synthesized silver nanoparticles were determined. Silver nanoparticles were tested for their antifungicidal efficacy against the most virulent isolate of the *Aspergillus flavus* fungus, JAM-JKB-BHA-GG20, and among the 10 different treatments, the treatment T6 [PDA + 1 ml of NP (19: 1)] + Pathogen was shown to be extremely significant (82.53%). TG-51 and GG-22 were found to be the most sensitive groundnut varieties after 5 and 10 days of LC-MS QTOF infection when 25 different groundnut varieties were screened using the most toxic *Aspergillus flavus* isolate JAM- JKB-BHA-GG20, respectively. In this research, the most susceptible groundnut cultivar, designated GG-22, was tested. Because less aflatoxin (1651.15 g.kg$^{-1}$) was observed, treatment T8 (Seed + Pathogen + 2 ml silver nanoparticles) was determined to be much more effective. The treated samples were examined by Inductively Coupled Plasma Mass Spectrometry for the detection of metal ions and the fungicide carbendazim. Ag particles (0.8 g/g$^{-1}$) and the fungicide carbendazim (0.025 g/g$^{-1}$) were found during Inductively Coupled Plasma Mass Spectrometry analysis below detectable levels. To protect plants against the invasion of fungal pathogens, environmentally friendly green silver nanoparticle antagonists with antifungal properties were able to prevent the synthesis of mycotoxin by up to 82.53%.

## Introduction

For the synthesis of nanoparticles (NPs), there are a number of chemical methods in the literature [1]. In these protocols, toxic chemicals are used, which have been a matter of great

database of NCBI with following accession number KU984480, KU984481, KU984482, KU984483, KU984484, KU984485, KU984486, KU984487, KU984488, KU984489, KU984490. Additional raw data, which is not part of the minimal data set for this study, is available from from the corresponding author on reasonable request.

**Funding:** The authors received no specific funding for this work.

**Competing interests:** The authors have declared that no competing interests exist.

concern for environmental reasons. As a result, researchers in the field of nanoscale material synthesis and assembly have been searching for alternatives in biological systems. Several biotechnological applications, such as microbial corrosion, bioremediation, biomineralization, and leaching, are highly dependent on the interactions between metal microbes [2,3]. Recently, the use of biological systems as a novel approach for the manufacture of metal nanoparticles has developed. Silver nanoparticles (AgNPs), being an important metal, have prospective uses in a wide range of fields, including medical diagnosis, catalysis, electronics, and antibacterial activity [4–7], and also have other unique properties such as anticoagulant, antidiabetic, and thrombolytic [8,9]. Among Me-NP, Ag-NPs have been known to have inhibitory and bactericidal effects [10,11]. Because of their huge specific surface area and high proportion of surface atoms, AgNPs are expected to exhibit more antibacterial activity than bulk silver metal. Resistance to commercially available antimicrobial drugs by pathogenic bacteria and fungi has become a serious problem in recent years [12]. Microbes, such as bacteria, molds, yeasts, and viruses, in living environment are often pathogenic and cause antimicrobial agents from natural and inorganic substances [10,13]. Among inorganic antimicrobial agents, silver has been used most extensively since ancient times to combat infections [14]. Compared to other microorganisms, fungi have the ability to produce a large amount of metabolites, which makes them more suitable for the production of nanoparticles [4,15]. It is well known that many microbes, both unicellular and multicellular, produce inorganic materials, intracellular or extracellular [2]. By reducing metal ions, microorganisms, including bacteria, yeast, and fungi, contribute significantly to the cleanup of hazardous metals and function as fascinating nanofactories [16]. These bacteria are excellent candidates for the manufacture of nanoparticles made of cadmium, gold, and silver (Ag-NP) [1,17,18]. For this reason, the present work has focused on the development of Ag-NPs using culture supernatant from antagonist bacteria and the evaluation of their antimicrobial activity against *Aspergillus* species producing aflatoxigenic, which cause serious problems about aflatoxin contamination, worldwide, in agricultural commodities, including groundnut. The characteristics of synthesized AgNPs were identified by using UV-vis spectroscopy, PSA, and zeta potential analysis, scanning electron microscopy, and Fourier transform infrared spectroscopy. We also tested the antifungal activity of AgNPs and standardized key factors involved in the process of synthesis of AgNPs.

## Materials and methods

### Synthesis and characterization of green silver nanoparticle antagonists from best antagonist

To obtain the cell supernatant for synthesis of silver nanoparticles from *Bacillus subtilis*, the bacterial strain was obtained from the Department of Microbiology of the Junagadh Agriculture University, Gujarat, India and was suspended aerobically in nitrate medium. Culture flasks were incubated on a rotator shaker at 120 rpm. Cell supernatant was collected after 24 hours by centrifugation at 6,000 rpm for 10 minutes at 6˚ C. For silver nanoparticle biosynthesis studies, the bacterial supernatant was inoculated in 250 ml Erlenmeyer flask containing 100 ml of sterile nutrient broth. Cultured flasks were incubated on a rotating shaker set at 200 rpm for 48 h at room temperature. After this, the culture was centrifuged at 12,000 rpm for 10 min. The biomass was separated by discarding the supernatant and used for the synthesis of silver nanoparticles. For this approximately 2 g of wet biomass containing *Bacillus subtilis* was resuspended in 100 ml of 1 mM AgNO$_3$ aqueous solution in a 250 ml Erlenmeyer flask, allowing the reduction of silver nanoparticles. All reaction mixtures were incubated on a rotating shaker (200 rpm) at room temperature for a period of 24 h in light. Heat-killed samples with AgNO$_3$ were also incubated along with experimental samples as a control. Visual observations were

conducted periodically to check for nanoparticle formation [19]. All media components and chemicals were analytical grade and obtained from Hi-Media Laboratories Ltd. Mumbai, India.

## Characterization of bacterial green silver nanoparticle antagonists

**Change in colour.** The colour change in the reaction mixture was recorded by visual observation. The change in supernatant colour from light yellow to dark brown indicated that the silver nanoparticles were synthesized as previously described [19].

**UV-vis spectral analysis.** Synthesized silver nanoparticles were confirmed by sampling the aqueous component at different time intervals, and the absorption maxima were scanned by UV-visible spectrophotometer at a wavelength of 300–700 nm on UV-visible spectrophotometer (Perkin Elmer Lambda 25 spectrophotometer), using deionized water as the reference [19].

**Fourier transform infrared (FTIR) spectroscopy.** Identification of the biomolecules associated with nanoparticle synthesis was performed using FTIR-8400S (Shimadzu). The dried silver nanoparticles were grinded with KBr pellets and measured in the wavelength range from 4000 to 400 cm$^{-1}$. Functional groups in silver nanoparticles were identified by FTIR by referring to the standard protocol [19].

**Particle Size Analyzer (PSA) and zeta potential analysis.** The particle size distribution and the Zeta potential analysis of synthesized nanoparticles were carried out by Microtrac S3500 PSA (Nanotrac Wave). The size distribution is determined on the basis of the dynamic scattering of red laser having wavelength 750 nm. Light is scattered by Brownian motion of the colloidal SNPs. Among the total percentage of the size distribution obtained, the value D-50, which is 50% size distribution, was taken into account. The z potential values suggest the stability of the synthesized nanoparticles as follows: Zeta potential value from 0 to ±5: Rapid coagulation or flocculation; Value from ±10 to ±30: Incipient instability; Value from ±30 to ±40: Moderate stability; Value from ±40 to ±60: Good stability; Value greater than ±61: Excellent stability.

**Scanning Electron Microscopy (SEM) analyses.** To reveal the shape and size, we applied AgNP scanning electron microscopy (SEM) analysis using the Zeiss Gemini SEM instrument. The samples were prepared onto an adhesive carbon tape on an aluminum stub. SEM is a direct technique used to detect and characterize nanoparticles. Scanning electron microscopy uses a high-energy electron beam, but the beam is scanned over the surface and backscattering of the electrons was observed [19].

## Bioefficacy of AgNPs

To optimize the effective dose of green AgNPs, we derived the best biocontrol for control of highly virulent and toxigenic *Aspergillus* strain below treatments were composed.

**In vitro assay.** The *in vitro* assay was performed on potato Dextrose Agar (PDA) growth medium treated with different concentrations (that is, 500 'μl, 1 ml, 2 ml, 3 ml and 4 ml) of AgNPs derived from the best antagonist strain of bacteria in conjunction with the control. AgNPs having different concentrations were poured into growth media prior to plating in a Petri dish ($90 \times 15$ mm). The medium containing silver nanoparticles was incubated at room temperature. After 48 h of incubation, uniform sized agar plugs (diameter, 8 mm) containing fungi were inoculated simultaneously in the center of each Petri dish containing AgNP, followed by incubation at $28 \pm 2°$ C for 5 and 10 days. All experiments were performed in triplicate, and the mean value along with standard deviation was recorded.

**Observations recorded.** After incubation of fungi on PDA medium containing silver nanoparticles, radial growth of fungal mycelium was recorded. Radial inhibition was

calculated when the growth of mycelia in the control plate reached the edge of the Petri dish. The following formula was used for the calculation of the inhibition rate (%):

$$\text{Inhibition rate } (\%) = \frac{R - r}{R}$$

Where, R is the radial growth of fungal mycelia on the control plate; r is the radial growth of fungal mycelia on the plate treated with AgNPs

## Collection of groundnut varieties

All methods were carried out in accordance with the relevant guidelines of the respective institute. The varieties of groundnut seeds used in the proposed research were developed and published by Junagadh Agriculture University, Gujarat, India, so no specific permission was required to obtain seeds for research. Additionally, since the work does not involve any endangered or protected plant species, no additional authorization is needed for collection of plant materials.

## Screening of susceptible groundnut variety and bioefficacy of green silver nanoparticle antagonists against aflatoxin-producing *Aspergillus* infections

Randomly, 10 grams of undamaged kernels were picked for the study of seed infection with Aspergillus flavus. The same procedure was followed for the selection of different varieties of groundnuts released for susceptibility and the aflatoxin estimation tests against the highly virulent and toxigenic *Aspergillus flavus* strain JAM-JKB-BHA-GG20 that was obtained from the Department of Microbiology of Junagadh Agriculture University, Gujarat, India. Aflatoxin was determined after 10 days of interval according to the LC-MS QTOF.

## Antagonist potentials of optimized AgNPs derived from the best antagonist with the most toxic aflatoxin-producing *Aspergillus* infected onto most susceptible groundnut seeds

To identify the effect of green AgNPs, best biocontrol for the control of highly virulent and toxigenic *Aspergillus* strains were composed below treatments.

**In vitro assay of pathogen inhibition by bacterial NPs.** The seeds of the most susceptible varieties released from groundnuts were collected from the Main Oil Seed Research Station, Junagadh Agricultural University, Gujarat. Randomly, 10 grams of undamaged kernels were picked for the study of seed infection with *Aspergillus flavus*. The same procedure was followed for the selection of susceptible groundnut varieties released for susceptibility and aflatoxin estimation test against the highly virulent and toxigenic *Aspergillus* strain. The aflatoxin was determined after 10 days of interval by LC-MS QTOF.

## Detection of Ag Ions Using Inductively Coupled Plasma Mass Spectrometry (ICPMS)

Inductively coupled plasma mass spectrometry (ICP-MS) can be used to find the total concentration of an element in a dissolved solution. ICP-MS works by nebulizing a solution, sending it through a spray chamber where droplets larger than 2 μm do not pass through, and then using argon plasma to ionize the atoms. The stream of ions goes through a series of cones where most of the ions do not pass through. The mass spectrometer of the instrument is kept under high vacuum along with the detector. A series of ion lenses focuses the ion stream on a mass spectrometer [20].

### Ethical statement

Each technique was used in accordance with the relevant institute-specific regulations. The varieties of groundnut seeds used in the proposed research were developed and published by Junagadh Agriculture University, Gujarat, India. Therefore, no specific permission was required to obtain seeds for research. Additionally, the work does not involve any endangered or protected plant species so no further authorization is needed for collection of plant material and utilization in the proposed research.

## Results

### Synthesis and characterization of green silver nanoparticle antagonists

The best biocontroller isolated was utilized for the green synthesis of nanoparticles. The bio-synthesized silver nanoparticles were characterized by using UV-vis spectroscopy, PSA, and Zeta Potential Analysis, Scanning Electron Microscopy and Fourier transform infrared spectroscopy.

**UV-vis spectroscopy.** A visible colour change of the solution was observed with the identified strain *Bacillus subtilis* when used for the synthesis of silver nanoparticles. The UV-vis spectroscopy was used to identify the silver nanoparticles. A characteristic broad peak of silver nanoparticles was observed in the UV-visible spectra at 430 nm. The blue peak was observed to shift in the absorption spectrum from 400 to 430 nm (Fig 1A and 1B).

### Scanning electron microscopy (SEM)

The nearly spherical form of the silver nanoparticles and diameters between 50 and 67 nm were verified by the SEM picture. The typical size of a nanoparticle is 62 nm (Fig 2).

### Particle Size Analysis (PSA) and zeta potential

Zetapotential measurements and particle size analyses were used to examine the quality of the nanodispersion. The majority of the particles in the dispersion were found to be smaller than 70 nm (average particle size), and the zeta potential was found to be -56.9 mV, indicating the presence of a stable nanodispersion (Fig 3A).

### Fourier transform infrared (FTIR)

To examine the reduction of $AgNO_3$ by the *Bacillus subtilis* bacterial isolate (JND-KHGn-29-A) culture supernatant, FTIR measurements were made to look for any interactions between silver salts and protein molecules that could be responsible for the reduction of Ag + ions and the stabilization of AgNP. The well-known electromagnetic fingerprints in the infrared part of the spectrum are created by the amide connections between amino acid residues in proteins. Weak bands between 1632.8 cm-1 and 1655.94 cm-1 were identified in the exudates of *Bacillus subtilis* as weak fingerprint areas of phenyl-ring substitution overtones. The band between 1673.3 and 1758.17 cm-1 that was detected is indicative of -C = O carbonyl stretching. The starching of peaks between 1239.31 $cm^{-1}$ and 1339.61 $cm^{-1}$ is due to the C-N amines bending vibration of the CN amines. While peaks between 1355.04 $cm^{-1}$ and 1466.91 $cm^{-1}$ are related to the C-H alkanes group's varied scissoring and bending vibrations of the C-H alkanes group. The carboxylic C-O group is responsible for the band at 1260.52 $cm^{-1}$ (Fig 3B).

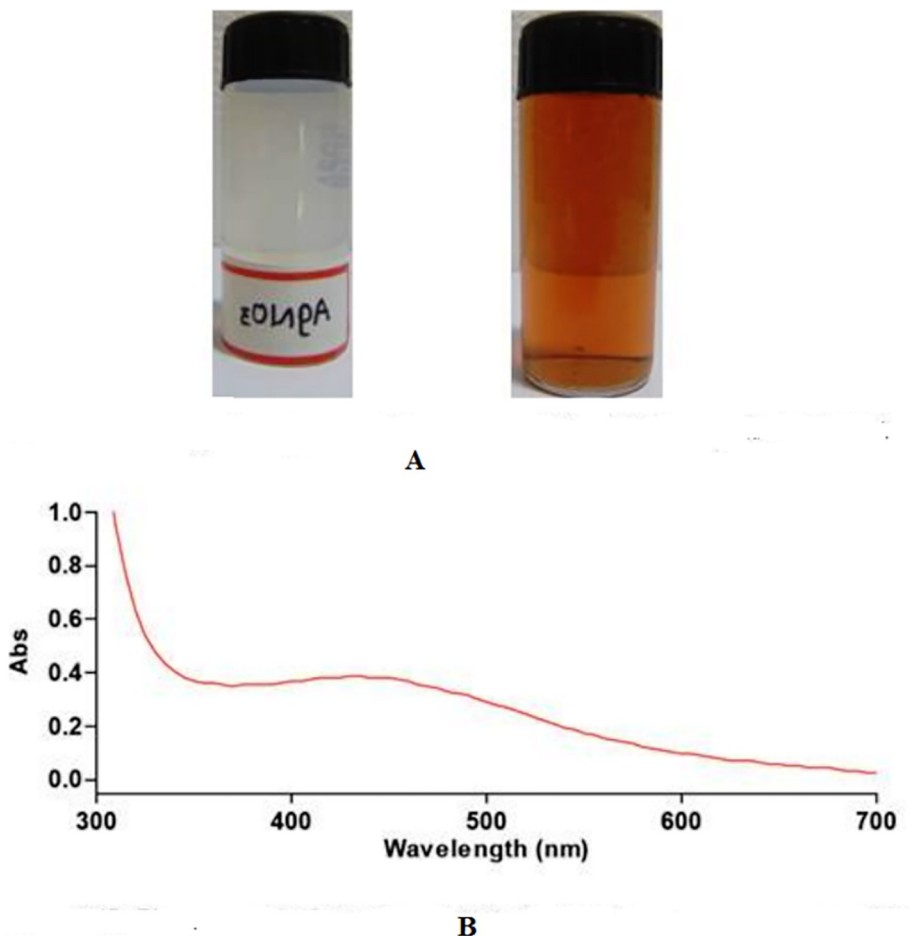

**Fig 1. Characterization of silver nanoparticles synthesized by *Bacillus subtilis*.** A: Monitoring of silver ions by change in color; B: The UV-VIS spectrum of nanoparticles.

## Antifungal assays

To determine the effective dosage of silver nanoparticles, tests were carried out on the most poisonous isolate of the *Aspergillus flavus* fungus, JAM-JKB-BHA-GG20. The most poisonous isolate of the *Aspergillus flavus* fungus, JAM-JKB-BHA-GG20, was shown to be inhibited (82.53%) by treatment T6 (PDA + 1ml NP (19: 1) + Pathogen), which was followed by treatment T3 (PDA (20 ml) +1 ml *Bacillus subtilis* ($2.7 \times 10^7$ cfu) + Pathogen) (Table 1 and S1 Fig in S1 File).

## Microscopic examination of antagonist interaction with test pathogen

To highlight the variations in fungal mycelium growth, the impact of the interaction between treatment T2: PDA (20 ml) + Fungus + *Bacillus subtilis* JND-KHGn-29-A (live antagonist) was shown under SEM. The results of a microscope examination showed that *Aspergillus flavus* JAM-JKB-BHA-GG20, the most dangerous strain of the fungus, was inhibited in its reproduction ability. Additionally, in T2 treatment, a potent antagonist formed over the pathogenic fungus's mycelia, forming a capsule-like spherical structure. Treatment T6: PDA + 1ml of NP (19: 1) + Pathogen destroyed the mycelia of the pathogenic fungus (Fig 4).

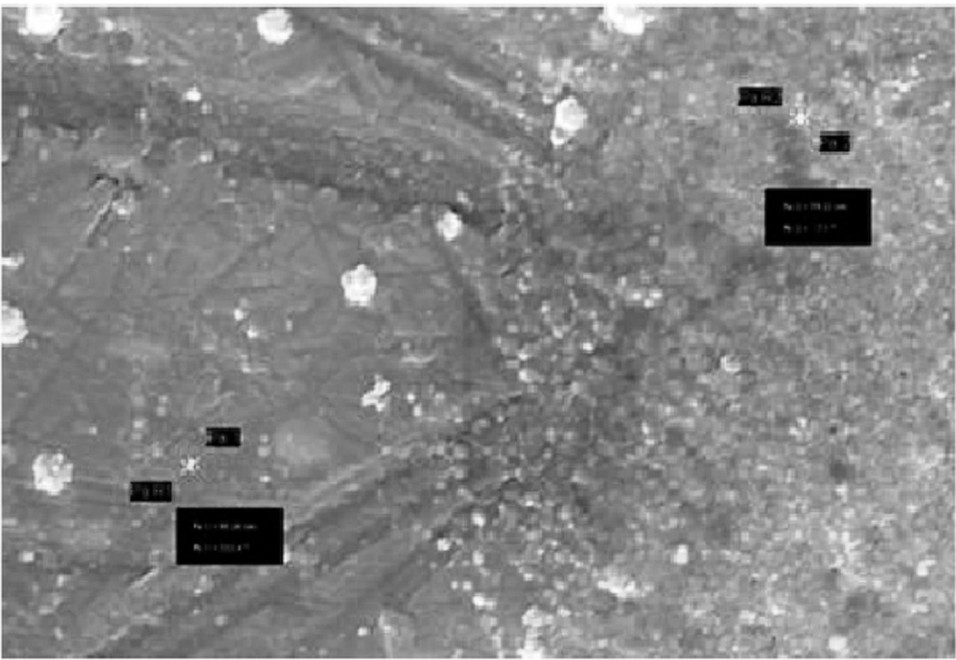

**Fig 2. Scanning Electron Microscope of silver nanoparticles synthesized by *Bacillus subtilis*.**

## Screening of susceptible groundnut variety and bio-efficacy of green silver nanoparticle antagonists

The *Aspergillus flavus* strain JAM-JKB-BHA-GG20 was used to infect all 25 cultivars, and aflatoxin generation and accumulation were observed after 5 and 10 days. In this investigation, the same fungus, JAM-JKB-BHA-GG20, was used to produce aflatoxin in 25 different types of groundnut beans. Table 2 lists the five resistant and five susceptible types that were found. Among the 25 JAU-released groundnut varieties, the ground nut variety GJG-19 (101 g.kg$^{-1}$) was found to be the most resistant; while the groundnut variety TG-51 (6111 g.kg$^{-1}$) was shown to be the most vulnerable. After 10 days, the 25 JAU groundnut variations showed that GJG-17 (551 g.kg$^{-1}$) was the most resistant variety and that GG-22 (35585 g kg-1) was the most vulnerable variety. (Tables 2 and 3 and Fig 5).

## Bio-efficacy of antagonists or green silver nanoparticle antagonists against aflatoxin producing *Aspergillus* infections in the most susceptible variety GG-22

**Screening of the most susceptible variety GG-22 with AgNPs.** The experiment was set up with 10 treatments to examine the effectiveness of bacterial nanoparticles in forming the aflatoxin content of the seed infected with *Aspergillus flavus* JAM-JKB-BHA-GG-20 in the most sensitive groundnut GG-22 (Table 4). After 7 days, LCMSQTOF was used to evaluate half of the samples, and the results showed that only the treatment T6, which used the pathogenic fungus JAM-JKB-BHA-GG20, produced aflatoxin. Treatment T8 (Seed+ Pathogen+2 m1 NP), which contained NP, was shown to be the most effective since less aflatoxin (1651.15 g.Kg$^{-1}$) was identified after pathogen infection. It shows that T8 therapy stops the generation of aflatoxin and the growth of the dangerous fungus *Aspergillus flavus* JAM-JKB-BHA-GG20. Because only groundnut seeds were used in treatment T6 (Seed + Pathogen* + 2 m1 water), a

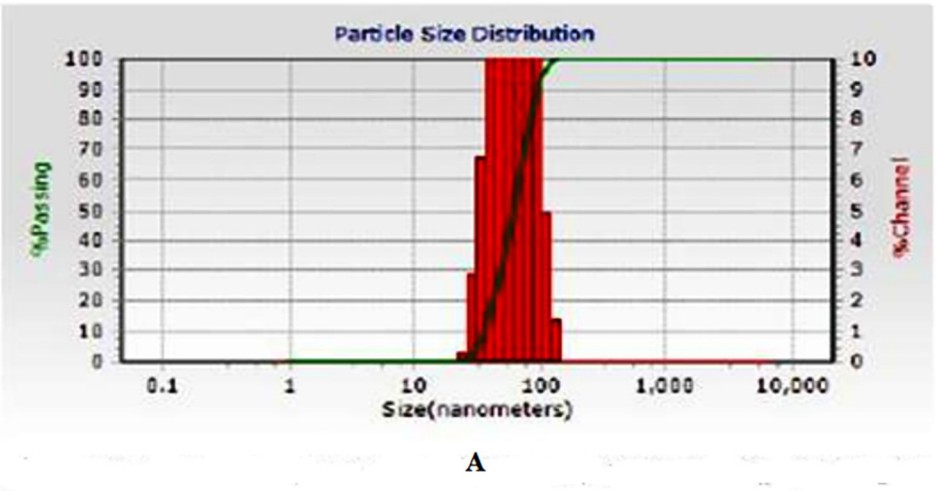

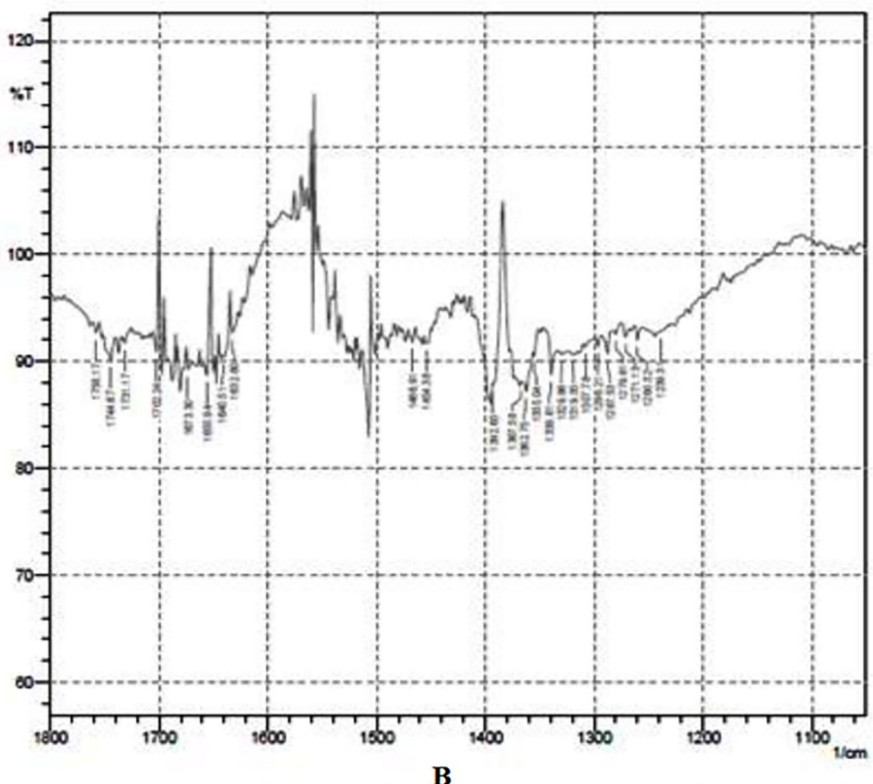

**Fig 3. Characterization of green synthesized bacterial AgNPs derived from best antagonists *Bacillus subtilis* JND-KHGn-29-A.** A: Particle size distribution of silver nanoparticles; B: FITR analysis of silver nanoparticles.

high quantity of aflatoxin (5388.82 g.kg$^{-1}$) was generated. The second halves of the remaining samples (treatments) were examined by ICPMS and GCMS for the detection of metal ions and fungicide. The outcome demonstrated the presence of the carbedazim fungicide carbedazim

**Table 1.  Percent growth inhibition of *Aspergillus flavus* by *Bacillus subtilis* synthesized nanoparticles.**

| Treatment No. | Treatment | % Growth inhibition of *A. flavus* |
|---|---|---|
| T1 | PDA (20 ml) + Pathogen* (Control) | 0.00 |
| T2 | PDA (20 ml) + Fungus + *B. subtilis* JND-KHGn-29-A (live antagonist)** | 57.05 |
| T3 | PDA (20 ml) + 1 ml *B. subtilis* ($2.7 \times 10^7$ cfu) + Pathogen | 72.53 |
| T4 | PDA + 200 µl nanoparticles (NPs) + Pathogen | 56.04 |
| T5 | PDA + 500 µl NPs + Pathogen | 67.03 |
| T6 | PDA + 1ml NPs (19:1) + Pathogen | 82.53 |
| T7 | PDA + 2ml NPs (18:2) + Pathogen | 67.03 |
| T8 | PDA + 3ml NPs (17:3) + Pathogen | 25.00 |
| T9 | PDA + 4ml NPs (16:4) + Pathogen | 0.00 |
| T10 | PDA + 5ml NPs (15:5) + Pathogen | 0.00 |
| | **S.E.M.±** | **0.53** |
| | **C.D. @ 5%** | **1.56** |
| | **C.V. %** | **2.32** |

* *Aspergillus flavus* JAM-JKB-BHA-GG20 (Isolate-3)—most toxic to produce aflatoxin

**NPs: nanoparticles from best antagonist *Bacillus subtilis* JND-KHGn-29-A; Cfu: Colony forming unit.

(0.025 g.g$^{-1}$) and Ag particles (0.8 g.g$^{-1}$) that were below MRL levels (Table 4 and Fig 6 and S2 Fig in S1 File).

## Discussion

The aim of the current work is to investigate the antifungal properties of silver nanoparticles generated from *Bacillus subtilis*, the most effective biocontrol agent. The potential antifungal activity of nanosized materials against *Aspergillus flavus*, a difficult groundnut pathogen, supports the idea that nanosizing produces antifungal activity and can serve as the best alternative to address the issue of such a significant pathogen for agriculture in the future. There are numerous synthetic antimicrobial substances that can be used in agriculture and medicine. However, the harmful effects of these agents on people and the environment, as well as widespread resistance to microbials, have led to an urgent search for microbial-based alternatives [21]. By opening up new pathways for the treatment of many diseases over the past three decades, innovations in the field of nanotechnology have revolutionized drug design and development [22]. In summary, the investigations performed as part of this thesis affirm the notion of nanosizing biocontrol microbes into biologically active substances. The highlights of the findings of these studies will be discussed in more detail.

Silver has long been known to exhibit strong toxicity to a wide range of 116 microorganisms [23] for these reasons silver-based compounds have been used extensively in many antimicrobial applications [24]. The aim of this study was the synthesis of AgNPs using a bacterial isolate, to characterize the synthesized nanoparticles and the evaluation of the antifungal activity of the synthesized AgNPs.

In the present study, the best biocontroller, i.e. *Bacillus subtilis*, was utilized for the green synthesis of nanoparticles, which was confirmed primarily as positive by changing the reaction mixture from the change of the blue peak in the absorption spectrum from 400 to 430 nm, indicating the production of silver nanoparticles (Ag+ to Ag0), which is in close agreement with the ranges reported by earlier researchers [25–27]. The shifting of blue in the AgNO$_3$ treated flask is attributed to the surface Plasmon resonance (SPR) that suggested the formation

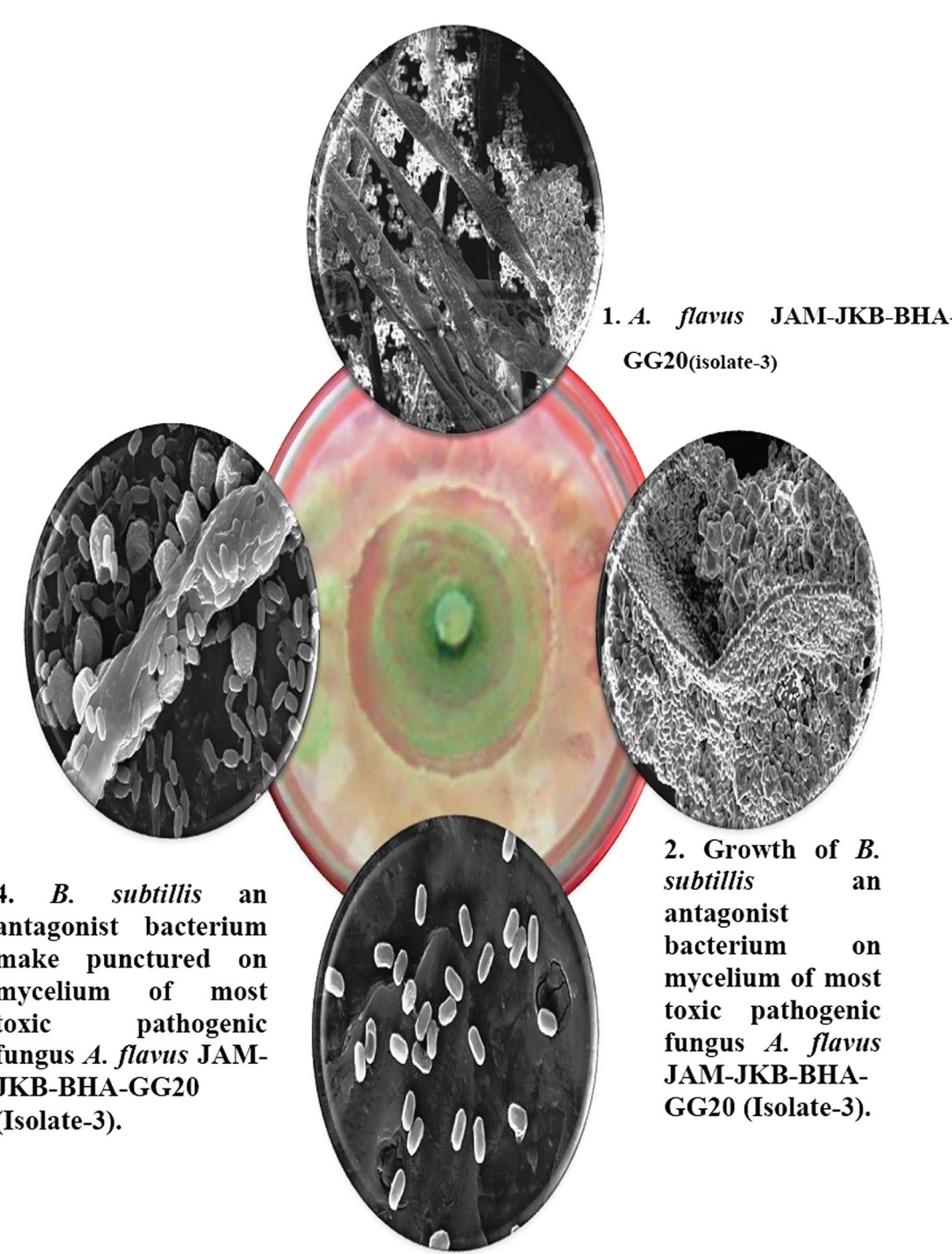

Fig 4. Scanning Electron Microscopic (SEM) image obtained from 8 day old fungal-bacterial antagonist interaction.

**Table 2. *In-vitro* screening of 25 groundnut varieties for aflatoxin production by most toxic and virulent *Aspergillus flavus* JAM-JKB-BHA-GG20 at 5 and 10 days of infections.**

| Duration | 5 days (Aflatoxin: µg.kg⁻¹) | | | | | 10 day (Aflatoxin: µg.kg⁻¹) | | | | |
|---|---|---|---|---|---|---|---|---|---|---|
| Variety | B1 | B2 | G1 | G2 | Total | B1 | B2 | G1 | G2 | Total |
| GG-2 | 53 | 51 | 29 | 99 | 232 | 955 | 462 | 701 | 618 | 2737 |
| GG-5 | 201 | 00 | 29 | 100 | 330 | 314 | 00 | 937 | 598 | 1848 |
| GG-6 | 42 | 00 | 30 | 99 | 171 | 611 | 00 | 792 | 00 | 1403 |
| GG-7 | 167 | 00 | 29 | 99 | 295 | 758 | 266 | 316 | 00 | 1341 |
| GG-8 | 43 | 64 | 31 | 97 | 236 | 2964 | 627 | 419 | 00 | 4010 |
| GG-9 | 42 | 00 | 30 | 97 | 169 | 3650 | 725 | 550 | 00 | 4925 |
| SB-XI | 64 | 00 | 00 | 00 | 101 | 3796 | 670 | 474 | 1070 | 6011 |
| GG-10 | 231 | 99 | 00 | 00 | 330 | 218 | 246 | 390 | 00 | 854 |
| GG-11 | 67 | 41 | 32 | 100 | 240 | 5674 | 1121 | 474 | 00 | 7268 |
| GG-12 | 3474 | 342 | 126 | 107 | 4049 | 2564 | 230 | 308 | 570 | 3672 |
| GG-13 | 102 | 43 | 32 | 103 | 280 | 12164 | 2030 | 312 | 00 | 14505 |
| GG-14 | 73 | 00 | 30 | 00 | 104 | 8820 | 1214 | 685 | 00 | 10719 |
| GJG-17 | 107 | 00 | 00 | 00 | 166 | 00 | 244 | 307 | 00 | 551 |
| GJG-18 | 74 | 41 | 00 | 00 | 115 | 00 | 3231 | 2487 | 00 | 23985 |
| GJG-19 | 33 | 68 | 00 | 00 | 101 | 26822 | 1234 | 598 | 00 | 28655 |
| GG-15 | 80 | 78 | 00 | 00 | 158 | 18725 | 2136 | 1022 | 00 | 21883 |
| GG-20 | 150 | 101 | 33 | 00 | 284 | 22523 | 365 | 530 | 00 | 23418 |
| GG-21 | 634 | 113 | 33 | 00 | 779 | 13369 | 2033 | 248 | 00 | 15650 |
| GG-22 | **214** | **103** | **31** | **00** | **348** | **32687** | **1460** | **1437** | **00** | **35585** |
| TG-26 | 54 | 60 | 32 | 00 | 146 | 4739 | 539 | 276 | 00 | 5553 |
| HPS-1 | 30 | 54 | 34 | 106 | 223 | 10663 | 1550 | 2959 | 00 | 15173 |
| TG-37 | 3380 | 501 | 146 | 00 | 4028 | 4090 | 356 | 244 | 00 | 4690 |
| TG-45 | 5008 | 459 | 136 | 00 | 5604 | 15339 | 2282 | 1132 | 00 | 18752 |
| TG-51 | 4924 | 890 | 297 | 00 | 6111 | 12306 | 1548 | 723 | 00 | 14577 |
| TPG-41 | 47 | 62 | 00 | 00 | 109 | 13313 | 1819 | 728 | 00 | 15859 |
| S.Em.± | **1.20** | **1.43** | **00** | **1.12** | **0.51** | **2.07** | **1.60** | **2.14** | **0.47** | **0.79** |
| C.D. @ 5% | **3.41** | **4.05** | **2.21** | **3.18** | **1.44** | **5.87** | **4.54** | **6.08** | **1.32** | **2.25** |
| C.V. % | **0.27** | **1.90** | **2.94** | **4.88** | **0.29** | **2.14** | **0.26** | **5.49** | **0.71** | **2.74** |

**Table 3. List of most susceptible and resistant groundnut varieties derived with infection of most toxic *Aspergillus flavus* JAM-JKB-BHA-GG20 by LCMS-Q-TOF.**

| 5 days | | | 10 days | | |
|---|---|---|---|---|---|
| Ground nut Variety | Total Aflatoxin (µg.kg⁻¹) | Pattern | Ground nut Variety | Total Aflatoxin (µg.kg⁻¹) | Pattern |
| GJG-19 | 101 | Tolerant Varieties | **GJG-17** | 551 | Tolerant Varieties |
| SB-XI | 101 | | **GG-10** | 854 | |
| GG-14 | 104 | | **GG-7** | 1341 | |
| TPG-41 | 109 | | **GG-6** | 1403.35 | |
| GJG-18 | 115 | | **GG-5** | 1848 | |
| GG-21 | 779 | Susceptible Varieties | **GG-15** | 21883 | Susceptible Varieties |
| TG-37 | 4028 | | **GG-20** | 23418 | |
| GG-12 | 4049 | | **GJG-18** | 23985 | |
| TG-45 | 5604 | | **GJG-19** | 28655 | |
| TG-51 | 6111 | | **GG-22** | 35585 | |

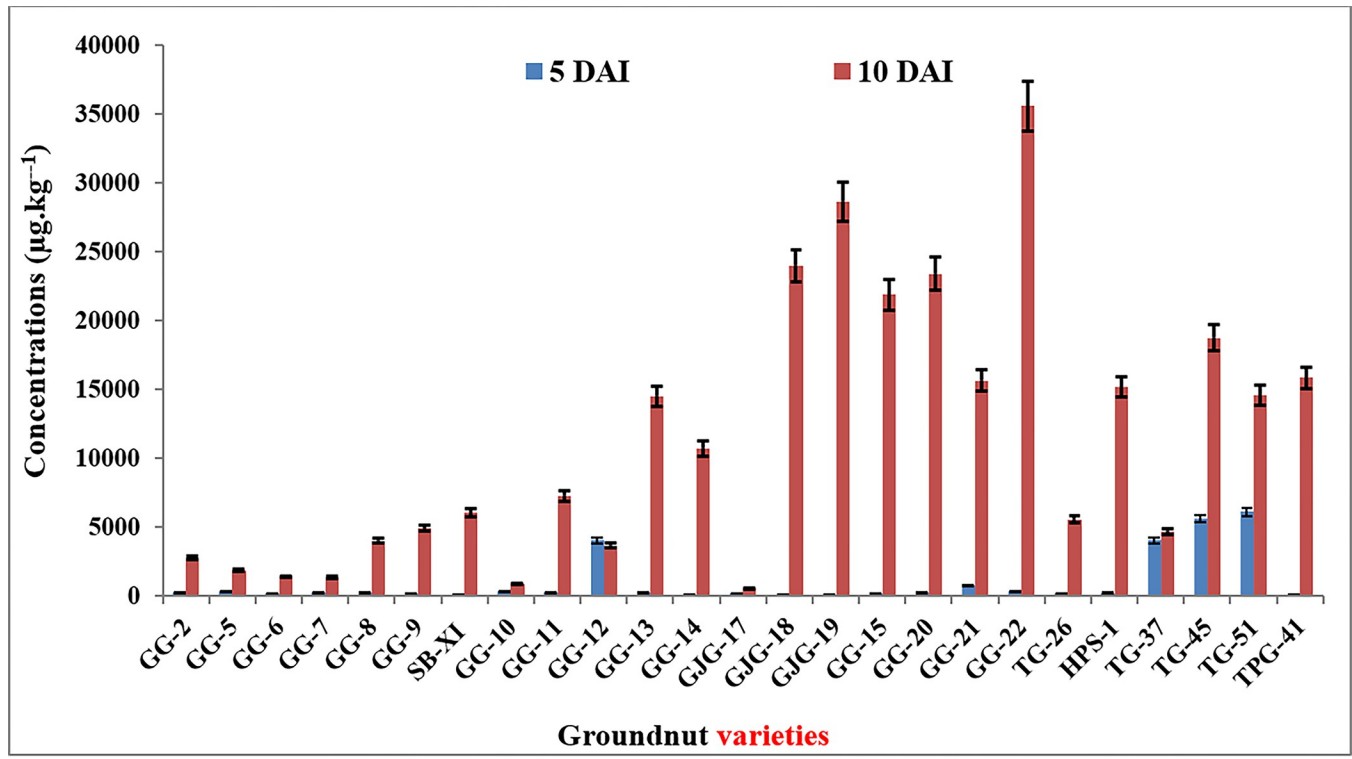

**Fig 5. Total aflatoxin production in 25 groundnut varieties by most toxic and virulent *Aspergillus flavus* JAM-JKB-BHA-GG20 after 5 and 10 days of infections.**

of AgNPs [28]. As such, it confirms the biosynthesis of silver nanoparticles using the best bio controller antagonistic bacterial culture i.e. *Bacillus subtilis* JND-KHGn-29-A.

Furthermore, the controversy over concluding that *Bacillus subtilis* produces AgNPs is obscured by the characterization of AgNPs using UV-vis spectroscopy, PSA, and Zeta

**Table 4. Efficacy of bacterial nanoparticles on production of seed aflatoxin content infected with *Aspergillus flavus* JAM-JKB-BHA-GG20 to most susceptible groundnut GG 22.**

| Treatment No. | Treatment | B1 µg.Kg⁻¹ | B2 µg.Kg⁻¹ | G1 µg.Kg⁻¹ | G2 µg.Kg⁻¹ | Total Aflatoxin µg.Kg⁻¹ |
|---|---|---|---|---|---|---|
| T1 | Seed + 2 ml Water (Control) | 0.00 | 0.00 | 0.00 | 0.00 | 0.00 |
| T2 | Seed + *B. subtilis*** in 2 ml Water | 0.00 | 0.00 | 0.00 | 0.00 | 0.00 |
| T3 | Seed + *B. subtilis* nanoparticles (NPs)*** 2 ml | 0.00 | 0.00 | 0.00 | 0.00 | 0.00 |
| T4 | Seed + AgNo3 in 2 ml Water | 0.00 | 0.00 | 0.00 | 0.00 | 0.00 |
| T5 | Seed + Fungicide in 2 ml Water | 0.00 | 0.00 | 0.00 | 0.00 | 0.00 |
| T6 | Seed + Pathogen* + 2 m1 Water | 5388.82 ± 2.08 | 0.00 | 0.00 | 0.00 | 5388.82 ± 2.08 |
| T7 | Seed + Pathogen + *B. subtilis* + 2 m1 Water | 4381.11 ± 3.21 | 0.00 | 0.00 | 0.00 | 4381.11 ± 3.21 |
| T8 | Seed + Pathogen +2 m1 NPs | 1651.15 ± 3.06 | 0.00 | 0.00 | 0.00 | 1651.15 ± 3.06 |
| T9 | Seed + Pathogen + 1 mM AgNo3 in 2 ml Water | 2403.08 ± 5.13 | 0.00 | 0.00 | 0.00 | 2403.08 ± 5.13 |
| T10 | Seed + Pathogen+ 1 mg carbedazim in 2 ml Water | 0.00 | 0.00 | 0.00 | 0.00 | 0.00 |

* *A. flavus* JAM-JKB-BHA-GG20—most toxic to produce aflatoxin

** *Bacillus subtilis* JND-KHGn-29-A live antagonist- best antagonist

***NPs: nanoparticles from best antagonist *B. subtilis* JND-KHGn-29-A; 1ug/g = 1000ug/kg.

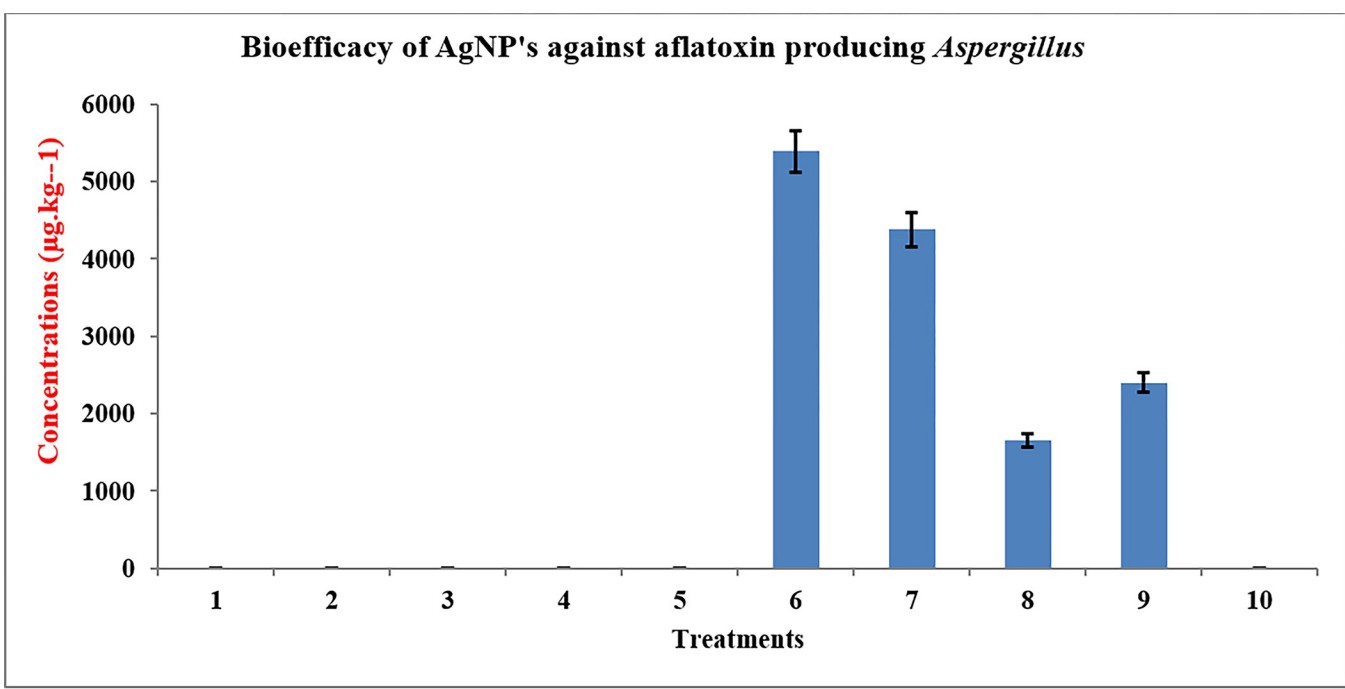

**Fig 6. Efficacy of bacterial nanoparticles on production of seed aflatoxin content infected with *Aspergillus flavus* JAM-JKB-BHA-GG20 to most susceptible groundnut GG 20.**

Potential Analysis, scanning electron microscopy, and Fourier transform infrared spectroscopy. In order to investigate the reduction of $AgNO_3$ by the culture supernatant of *Bacillus subtilis* bacterial isolate no. 1 (JND-KHGn-29-A), FTIR measurements were made to look for any interactions between silver salts and protein molecules that could be responsible for the reduction of Ag + ions and the stabilization of AgNP. The amide connections of proteins between amino acid residues provide the well-recognized electromagnetic fingerprints in the infrared range. The overall FTIR pattern confirms the presence of proteins in the synthesized nanoparticles. The free amine and carbonyl groups present in the bacterial protein could possibly perform the function for the formation and stabilization of silver nanoparticles [29,30]. The biosynthesis of silver nanoparticles using *Bacillus* sp. with a size range of 50–100 nm was reported by Safaa et al. [31], Vithiya et al. [32], Vidhya et al. [33] which support our findings. The findings of Sinha et al. [34] to identified possible interactions between silver salts and protein molecules, which could explain the reduction of Ag + ions and stabilization of AgNPs, were similar to our findings.

The existence of a plasmon absorption band at around 430 nm, which is typical of silver nanoparticles, in the absorption spectra of the silver nanoparticles, as illustrated, showed their creation. The surface of a metal resembles plasma with unbound electrons in the conduction band and positively charged nuclei. Surface Near the surface of the nanoparticles, plasmon resonance is a collective excitation of the conduction band electrons. Electrons are limited to specific vibrational modes by the particle's size and shape. The large peak at 431 nm is caused by surface plasmons, which are unique to Ag nanoparticles and result from the collective oscillations of valence electrons in the electromagnetic field of incoming light [35].

The 25 varieties were infected with *Aspergillus flavus* JAM-JKB-BHA-GG20 and aflatoxin production and accumulations were identified at 5 and 10 days of infection. Aflatoxin extraction and LCMS-Q TOF quantification were acquired. In this study, the 25 grounds varieties

apparently behaved for aflatoxin production infected with the same fungus. In this study, the five resistant and five susceptible varieties were identified. It is intriguing that certain varieties of peanuts support low toxin production, whereas other varieties support maximal production. This difference is possibly related to certain basic biochemical characteristics such as protein [36] or possibly vitamin E [37] in addition to cultivar practices. From the point of view of prevention, the challenge, therefore, appears to be to identify peanut genotypes that will support minimal toxin production by a number of fungal isolates of *Aspergillus flavus* and *A. Parasiticus*. *A. Parasiticus* is well recognized to be powerfully toxigenic. Shencong et al. [38] demonstrated a sensitive and selective analytical method for the determination of the residues of aflatoxin G1, G2, B1, and B2 in groundnuts using the Agilent G6410AA Triple Quadrupole Mass Spectrometer LC / MS. Simple sample preparation techniques are used in this procedure; then LC/MS/MS. The detection thresholds for all aflatoxins in cereals were less than 1 ng.mL$^{-1}$.

To evaluate the efficacy of bacterial nanoparticles in the production of seed aflatoxin content inoculated with *Aspergillus flavus* JAM-JKB-BHA-GG20 to the most susceptible groundnut GG-22, the experiment was served with 10 treatments. After 7 days, half of the samples were analyzed by LCMS-Q-TOF, which caused aflatoxin to only be produced for treatment T6. Among the various treatments, treatment T8, (Seed + Pathogen + 2 m1 NP) containing NP, was less significant, as less aflatoxin concentration of aflatoxin was detected (1651.15 μg. Kg$^{-1}$) upon pathogen infection. Safaa et al. [31] revealed that AgNPs can be effectively against various pathogenic plant fungi. *In vitro* Petri dish assays indicated a significant effect on the colony formation of these two pathogens. Effective concentrations of AgNPs inhibited colony formation of *B. sorokiniana* and *M. grisea*. Growth chamber inoculation tests provided additional evidence that the two fungi that cause these diseases in perennial ryegrass (*Lolium perenne*) had been greatly reduced by AgNPs. Sang et al. [39] reported fungicidal properties of silver nanoparticles against pathogenic plant fungi. Eighteen different pathogenic plant fungi were treated with AgNPs on plates of potato dextrose agar (PDA), malt extract agar, and corn meal agar. The results indicated that AgNPs possess antifungal properties against these plant pathogens at various levels and showed that the most significant inhibition of plant pathogenic fungi was observed in PDA, which supports our findings.

The majority of chemical and physical techniques for producing nanosilver are very costly and require the use of dangerous, poisonous compounds that might be harmful to the environment or human health. There is a need for an ecologically and financially viable method of synthesizing silver nanoparticles, since it is an inevitable truth that these particles must be handled by people and made more affordable for successful use. The search for such a technique has necessitated the development of biomimetic silver nanoparticle synthesis, in which biological techniques are employed to create the particles, and their application in large-scale manufacturing has been shown to be economically advantageous. As such, the synthesized AgNPs exhibited remarkable antifungal activity against *Aspergillus flavus* regardless of their drug resistance mechanisms and could also be considered as a potential antifungal agent. In addition to overcoming resistance and being less expensive than traditional antibiotics, fungicidal activity has demonstrated that AgNPs kill germs at low concentrations (units of ppm) without revealing acute harmful effects on human cells.

## Supporting information

**S1 File. S1 and S2 Figs: Detailed information on the groundnut varieties used in the proposed study and results of screening analysis.**
(PDF)

## Author Contributions

**Conceptualization:** Sunil Tulshiram Hajare.

**Data curation:** Sunil Tulshiram Hajare, Vijay Upadhye.

**Formal analysis:** Mukesh Soni, Krushna Kant Prajapati.

**Investigation:** Achyut Ashokrao Bharose.

**Resources:** Gajera H. P., Mukesh Soni.

**Software:** Krushna Kant Prajapati.

**Supervision:** Gajera H. P.

**Validation:** Vijay Upadhye.

**Visualization:** Suresh Chandra Singh.

**Writing – original draft:** Sunil Tulshiram Hajare.

**Writing – review & editing:** Sunil Tulshiram Hajare.

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
