## [Decision Letter · Decision Letter 0]

30 Oct 2023

PONE-D-23-27369Bacteria-mediated green synthesis of nano-silver particles and their anti-fungal potentials against Aspergillus flavusPLOS ONE

Dear Dr. Hajare,

Thank you for submitting your manuscript to PLOS ONE. After careful consideration, we feel that it has merit but does not fully meet PLOS ONE’s publication criteria as it currently stands. Therefore, we invite you to submit a revised version of the manuscript that addresses the points raised during the review process.

We look forward to receiving your revised manuscript.

Kind regards,

Arumugam Sundaramanickam, PhD

Academic Editor

PLOS ONE

Journal Requirements:

https://journals.asm.org/doi/pdf/10.1128/am.25.2.319-321.1973

In your revision ensure you cite all your sources (including your own works), and quote or rephrase any duplicated text outside the methods section. Further consideration is dependent on these concerns being addressed.

"NO"

**Additional Editor Comments:**

ACADEMIC EDITOR:I have carefully reviewed your submission and would like to offer some feedback based on the Submission Guidelines. It is essential to ensure that your manuscript complies with the provided guidelines to enhance the quality and suitability of your work for publication.Please find below the necessary modifications:Submission Guidelines Compliance: Ensure that your manuscript aligns with the provided Submission Guidelines. This includes adhering to formatting, style, and length requirements. Particularly the Materials and Methods should moved before the results section Methodology Section Revision: The methodology section requires major revisions. Please consider the following: Sampling Procedure Details: Provide a comprehensive description of the sampling procedure, How the bacterial strain  *Bacillus subtilis *was  isolated? include the sampling methods. This will help readers understand about your research. Proper References: Ensure that all references in the methodology section are accurate and appropriately cited. This is crucial for establishing the credibility of your research. https://journals.plos.org/plosone/s/submission-guidelines#loc-manuscript-organization

Reviewers' comments:

Reviewer's Responses to Questions

**Comments to the Author**

1. Is the manuscript technically sound, and do the data support the conclusions?

Reviewer #1: Partly

Reviewer #2: No

2. Has the statistical analysis been performed appropriately and rigorously? 

Reviewer #1: No

Reviewer #2: No

3. Have the authors made all data underlying the findings in their manuscript fully available?

Reviewer #1: Yes

Reviewer #2: No

4. Is the manuscript presented in an intelligible fashion and written in standard English?

Reviewer #1: Yes

Reviewer #2: No

5. Review Comments to the Author

Reviewer #1: 1. In table 4, Most susceptible groundnut GG 20 has mentioned but in your abstract “The GG-22 was identified as most susceptible groundnut variety and was served in further study.” has mentioned. Same you have mentioned in figure 6 that “Efficacy of bacterial nanoparticles on production of seed aflatoxin content infected with A. flavus JAM-JKB-BHA-GG20 to most susceptible groundnut GG 20”.

2. Screening of susceptible groundnut variety and bio-efficacy of antagonists green silver Nanoparticles.

After 5 days, the ground nut variety GJG-19 (101 μg.kg-1) was identified as most resistant variety whereas, ground nut variety TG-51 (6111 μg.kg-1) was identified as most susceptible variety among all 25 JAU released groundnut varieties. After 10 days, the groundnut variety GG-22 (35585 μg.kg-1) was identified as most susceptible variety whereas, GJG-17 (551 μg.kg-1) was identified as most resistant variety among 25 JAU realized groundnut varieties. The GG-22 was selected as most susceptible groundnut variety and used in further study.

Why they have used G22 for further studies not TG-51? Scientific Explanation?

3. From where you have isolated Bacillus subtilis and Aspergillus flavus?

4. Synthesis and characterization of green silver nanoparticles from best antagonist

As you have mentioned there, the cultured flasks were incubated for 48 hours at room temperature on a revolving shaker set at 200 rpm. The culture was then centrifuged for 10 minutes at 12,000 rpm. What do you mean by culture flask?

5. For this reason, the present work has been focused on the development of Ag-NPs using culture supernatant of antagonist bacteria and the evaluation of their antimicrobial activity against Aspergillus species producing aflatoxigenic, which cause serious problems about aflatoxin contamination, worldwide in agricultural commodities including groundnut.

What do you mean by cultured supernatant antagonist?

6. In figure 5, X-axis labelling “Ground varities” spelling has wrong.

Reviewer #2: English-wise, the manuscript is poorly written and needs major careful revision.

The manuscript suffers from poor information about the mechanism of synthesis of nanoparticles.

How did the bacteria synthesize the nanoparticles and stabilize NPs? What analysis proved this statement? What is the proposed mechanism?

The study lacks statistical analyses to show the significance of the results.

The discussion part is written poorly, and the available data is much more; the authors need to improve the study with more literature review.

What was the yield of biosynthesis of silver nanoparticles?

The authors are suggested including a paragraph to mention the different physical, chemical, and biological techniques for synthesis of inorganic nanoparticles and explain the superiorities of microbial synthesis of nanoparticles over traditional physical and chemical synthesis.

The abbreviations should come later than the actual words and provide the details when used for first time in the manuscript.

There are lots of spelling and grammatical mistakes that occur throughout the manuscript, for example interection, grinded, nanoparticals etc..; authors should check them carefully and need to proofread the paper by English professionals.

None of the characterization figures are clear and unsuitable; please improve their quality.

The authors must put each chemical purity or grade and make in the material and methods section.

In the SEM image, I couldn’t see any nanoparticles; I strongly recommend authors should improve the image quality. There are no measurement details in the SEM micrograph, which is very important in the analysis.

In the DLS image complete peak should be visible there is no clear data found

FTIR graph not clear provide images with clarity; Standard scanning range of FTIR is 4000-400 cm-1; the author mentioned only 1800-1100, what is the reason behind? Please provide complete scan data and discuss the functional groups.

Provide proper subheadings, change improper headings like ‘By color change’

Avoid abbreviations in the abstract part

The word ‘Anti-fungicidal activity’ should replace by ‘Anti-fungal activity’

Figure (4) is badly shown, I suggest separating that figure into five to make things clear;

Centre of the cultural plates showed visual Aspergillus growth, how can you claim that NPs inhibit the fungus more than 70%.

How the silver nanoparticles are better than the existing methods, since silver is an expensive material? How it can be cost-effective in large scale production?

What is the purification of nanoparticles in the study?

How the bacterial strains were discarded after synthesis of nanoparticles?

What is the product drying process in the study?

Invitro fungicidal activity of different concentrations of silver nanoparticles can be included (Culture plates) as visual evidence.

6. PLOS authors have the option to publish the peer review history of their article (what does this mean?). If published, this will include your full peer review and any attached files.

Reviewer #1: **Yes: **Surabhi Awasthi

Reviewer #2: No

---

## [Author Response · Author response to Decision Letter 0]

7 Dec 2023

Response to Reviewers Comments

Academic Editor

1. Financial statement has been modified in the revised manuscript.

2. Data availability statement has been modified in the revised manuscript.

3. Captions for the supporting information have been provided in the revised paper.

4. Materials and method section has been moved before the result section in the revised version of paper.

5. Brief description for isolation of B. subtilis has been added in the revised draft of paper.

6. All references are properly cited in the revised paper.

7. Statement for replicates has been included in the revised paper.

8. Since, the manuscript involves most of the analysis part through various instrumentation it does not required appropriately and rigorously statistical analysis.

Reviewer 1

1. We apologize for the typographical errors in Table 4 and Figure 6. The same has been corrected in the revised paper by mentioning G22 as susceptible variety.

2. We agree with the learn reviewer that TG-51 and GG-22 were found to be the most sensitive groundnut varieties after 5 and 10 days of infection by LC-MS QTOF when 25 different groundnut varieties were screened using the most toxic A. flavus isolate JAM- JKB-BHA-GG20 respectively. The most susceptible groundnut cultivar, designated as GG-22, was tested in this research. Because less aflatoxin (1651.15 g.kg-1) was observed, treatment T8 (Seed + Pathogen + 2 m1 AgNPs) was determined to be much more effective. Hence, G-22 was designated as most susceptible varieties and used for further studies.

3. The details of isolation of Bacillus and Aspergillus species have been provided in the revised paper.

4. The culture flask means the conical flasks which were used for experiments.

5. Culture supernatant antagonist refers to a substance obtained from the liquid portion of bacterial culture which has antagonist effect by inhibiting the growth of Aspergillus which indicates the interaction and dynamics within the microbial communities.

6. Spelling of Groundnut varieties has been corrected in Figure 5.

Reviewer 2

1. We agree with the learned reviewer and the quality of English language have been improved by taking the help of person who is expert in English language. 

2. As suggested by learned reviewer a brief description for synthesis of silver particles have been inserted in the revised manuscript.

3. Bacillus subtilis posses the enzyme nitrate reductase enzyme which helps in reduction and stabilization of silver nanoparticles. This is common mechanism. Synthesis of silver nanoparticles was confirmed by Utilizing UV-Visible spectroscopy, PSA and Zeta Potential Analysis, Scanning Electron Microscopy, and Fourier Transform Infrared Spectroscopy. The same has been clearly mentioned in the revised paper.

4. Since, the manuscript involves most of the analysis part through various instrumentation it does not required appropriately and rigorously statistical analysis. However, all the important statistical analysis was utilized to prove the significance of study. The statistical analysis are in clearly indicates in respective tables and figures.

5. As recommended by reviewer discussion has been modified in the revised paper.

6. The yield of nanoparticles was in the range of 50 to 67 nm. 

7. We agree with the learn reviewer and the paragraph superiorities of microbial synthesis of nanoparticles over traditional physical and chemical synthesis has been added in the revised manuscript.

8. Abbreviations are put later than the actual word in the revised paper as recommended by learn reviewer.

9. All spellings, grammatical and typo errors are corrected in the revised paper by taking the help of person who is native in English.

10. The purity and grade of chemical has been added in the revised paper.

11. We agree with the reviewer however, all the figures are modified as per the PACE software tools to meet the requirement of journals and also improved the quality of figures.

12. We totally agree with the learn reviewer however, FITR is shown in the range of 1800-1100 to demonstrate the best possible results by referring standard references.

13. Sub-headings are modified in the revised paper.

14. Abbreviations are removed from abstract section in revised paper.

25. The word ‘Anti-fungicidal activity’ was replace by ‘Anti-fungal activity’ in revised paper.

16. Figure 4 shows the correlation among each other and therefore it is shown together and modified through PACE to meet the journal requirements.

17. We agree with the learn reviewer however, no growth was observed for fungus across the centre well which confirmed that the growth was inhibited by more than 70%.

18. Justification for cost effective of silver nanoparticles has been provided in the revised paper under discussion section.

19. Culture was centrifuged to remove the impurities in order to obtain pure nanoparticles and is clearly mentioned in paper.

20. Bacterial strains were discarded by autoclaving them by referring safety laboratory protocols.

21. Drying process is not included in this study as we follow other standard reference published previously and hence is not mentioned in the paper.

22. We agree that the In-vitro fungicidal activity of different concentrations of silver nanoparticles can be included (Culture plates) as visual evidence and is provided as supplementary file for clear understanding. Also we have to follow journal guidelines to reduce number of tables and figures.

Other Corrections

1. All the typographical errors, minor corrections, etc. are highlighted with the track change in the revised manuscript.

---

## [Decision Letter · Decision Letter 1]

15 Jan 2024

Bacteria-mediated green synthesis of nano-silver particles and their anti-fungal potentials against Aspergillus flavus

PONE-D-23-27369R1

Dear Dr. Hajare,

We’re pleased to inform you that your manuscript has been judged scientifically suitable for publication and will be formally accepted for publication once it meets all outstanding technical requirements.

Kind regards,

Arumugam Sundaramanickam, PhD

Academic Editor

PLOS ONE

Additional Editor Comments (optional):

Reviewers' comments:

Reviewer's Responses to Questions

**Comments to the Author**

1. If the authors have adequately addressed your comments raised in a previous round of review and you feel that this manuscript is now acceptable for publication, you may indicate that here to bypass the “Comments to the Author” section, enter your conflict of interest statement in the “Confidential to Editor” section, and submit your "Accept" recommendation.

Reviewer #2: All comments have been addressed

2. Is the manuscript technically sound, and do the data support the conclusions?

Reviewer #2: Partly

3. Has the statistical analysis been performed appropriately and rigorously? 

Reviewer #2: No

4. Have the authors made all data underlying the findings in their manuscript fully available?

Reviewer #2: Yes

5. Is the manuscript presented in an intelligible fashion and written in standard English?

Reviewer #2: Yes

6. Review Comments to the Author

Reviewer #2: I would like to appreciate author’s effort, it is highly commendable. Authors have addressed all the comments raised by me. The manuscript is now acceptable for publication

7. PLOS authors have the option to publish the peer review history of their article (what does this mean?). If published, this will include your full peer review and any attached files.

Reviewer #2: No

---

## [Editor Report · Acceptance letter]

15 Mar 2024

PONE-D-23-27369R1 

PLOS ONE

Dear Dr. Hajare, 

I'm pleased to inform you that your manuscript has been deemed suitable for publication in PLOS ONE. Congratulations! Your manuscript is now being handed over to our production team.

Kind regards, 

on behalf of

Professor Arumugam Sundaramanickam 

Academic Editor

PLOS ONE